# Short Report: Early genomic detection of SARS-CoV-2 P.1 variant in Northeast Brazil

**Stephane Tosta**[1,2☯], **Marta Giovanetti**[1,3☯]*, **Vanessa Brandão Nardy**[2], **Luciana Reboredo de Oliveira da Silva**[2], **Marcela Kelly Astete Gómez**[2], **Jaqueline Gomes Lima**[2], **Cristiane Wanderley Cardoso**[4], **Tarcisio Oliveira Silva**[5], **Marcia São Pedro Leal de Souza**[6], **Pedro Henrique Presta Dia**[7], **Vagner Fonseca**[1,8,9], **Tulio de Oliveira**[9], **José Lourenço**[10], **Luiz Carlos Junior Alcantara**[1,3]*, **Felicidade Pereira**[2☯]*, **Arabela Leal**[2]*

1 Laboratório de Genética Celular e Molecular, Instituto de Ciências Biológicas, Universidade Federal de Minas Gerais, Belo Horizonte, Minas Gerais, Brazil, 2 Laboratório Central de Saúde Pública da Bahia–LACEN-BA, Salvador, Bahia, Brazil, 3 Laboratório de Flavivírus, Instituto Oswaldo Cruz, Fundação Oswaldo Cruz, Rio de Janeiro, Brazil, 4 Secretaria Municipal de Saúde de Salvador, Salvador, Bahia, Brazil, 5 Secretaria Municipal de Saúde de Irecê, Irecê, Bahia, Brazil, 6 Diretoria de Vigilância Epidemiológica do Estado da Bahia (DIVEP), Salvador, Bahia, Brazil, 7 Centro e Informações Estratégicas de Vigilância em Saúde do estado da Bahia (CIEVS), Salvador, Bahia, Brazil, 8 Coordenação Geral dos Laboratórios de Saúde Pública/Secretaria de Vigilância em Saúde, Ministério da Saúde, (CGLAB/SVS-MS) Brasília, Distrito Federal, Brazil, 9 KwaZulu-Natal Research Innovation and Sequencing Platform (KRISP), School of Laboratory Medicine and Medical Sciences, College of Health Sciences, University of KwaZulu-Natal, Durban, South Africa, 10 University of Oxford, department of Zoology, Oxford, United Kingdom

☯ These authors contributed equally to this work.
* giovanetti.marta@gmail.com (MG); luiz.alcantara@ioc.fiocruz.br (LCJA); felicidade.pereira@saude.ba.gov.br (FP); arabelaleal@gmail.com (AL)

## Abstract

Tracking the spread of SARS-CoV-2 variants of concern is crucial to inform public health efforts and control the ongoing pandemic. Here, we report genetic evidence for circulation of the P.1 variant in Northeast Brazil. We advocate for increased active surveillance to ensure adequate control of this variant throughout the country.

## Author summary

In recent months' variants of SARS-CoV-2 that have more mutations on the Spike protein has brought concern all over the world. These have been called 'variants of concern' (VOC) as it has been suggested that their genome mutations might impact transmission, immune control, and virulence. The P.1 variant, also known as 20J/501Y.V3, was first identified in travelers from Brazil during routine airport screening in Tokyo, Japan, in early January 2021. This VOC has 17 amino acid changes, ten of which are in its spike protein, including three designated to be of particular concern: N501Y, E484K and K417T. Since its first detection, despite it has presented sustained transmission worldwide, much is still unknown about its circulation into Brazilian regions. Here, through an active monitoring conducted by public health authorities of the Bahia state (Northeast Brazil), we report genetic evidence for circulation of the P.1 variant into the state. Our findings reinforce that continued genomic surveillance strategies are needed to assist in the monitoring

**Data Availability Statement:** Newly generated SARS-CoV-2 sequences have been deposited in

GISAID under accession numbers EPI_ISL_1067728 to EPI_ISL_1067738 and are already available at: https://www.gisaid.org/.

**Funding:** This work was financed by Laboratório Central de Saúde Pública da Bahia (LACEN-BA) and Secretaria da Saúde do Estado da Bahia (SESAB). ST is supported by the Coordenação de Aperfeiçoamento de Pessoal de Nível Superior – Brasil (CAPES) – Finance Code 001. MG is supported by Fundação de Amparo à Pesquisa do Estado do Rio de Janeiro (FAPERJ). JL is supported by a lectureship from the Department of Zoology, University of Oxford. This work was support in part through National Institutes of Health USA grant U01 AI151698 for the United World Arbovirus Research Network (UWARN). The funders had no role in study design, data collection and analysis, decision to publish, or preparation of the manuscript.

**Competing interests:** The authors have declared that no competing interests exist.

and understanding of the circulating and co-circulating SARS-CoV-2 variants, which might help to attenuate their public health impact worldwide.

## Introduction

Since the emergence of the severe acute respiratory syndrome coronavirus-2 (SARS-CoV-2) in 2019, the combination between the unprecedented number of cases and more than 500K genomes generated has allowed the identification of hundreds of circulating genetic variants during the pandemic [1]. Currently, three variants (B.1.1.7 or VOC202012/01, B.1.351 or 20H/501Y.V2 and P.1) carrying several mutations in the receptor-binding domain (RBD) of the spike (S) protein, raise concerns about their potential to shift the dynamics and public health impact of the pandemic [2–5]. They appear potentially associated with (i) increased transmissibility, (ii) propensity for re-infection, (iii) escape from neutralizing antibodies, and (iv) increased affinity for the human ACE2 receptor [6–8].

First identified in January 2021 in travelers from the Amazonas state (North of Brazil) who arrived in Japan, the P.1 variant (alias of B.1.1.28.1) [9], harbors a constellation of 17 unique mutations, including three in the receptor binding domain of the spike protein (K417T, E484K, and N501Y). It thus immediately raised concerns to public health authorities over the risk of its unknown potential of faster spreading and/or worsening of coronavirus disease (COVID-19) clinical outcomes. In view of its rapid spread in Brazil and elsewhere [1,4,5] the public health authorities of the Bahia state (Northeast Brazil) conducted an active monitoring for a rapid detection of this variant in the state.

Here, we report genetic evidence of the circulation of the P.1 variant in Bahia, by generating 11 SARS-CoV-2 complete genomes from travelers returning from the Amazonas state (North of Brazil).

## Material and methods

### Ethics statement

This research was approved by the Ethics Review Committee of the Pan American Health Organization (PAHOERC.0344.01) and the Federal University of Minas Gerais (CEP/CAAE: 32912820.6.1001.5149). The availability of these samples for research purposes during outbreaks of national concern is allowed to the terms of the 510/2016 Resolution of the National Ethical Committee for Research–Brazilian Ministry of Health (CONEP—Comissão Nacional de Ética em Pesquisa, Ministério da Saúde), that authorize, without the necessity of an informed consent, the use of clinical samples collected in the Brazilian Central Public Health Laboratories to accelerate knowledge building and contribute to surveillance and outbreak response.

### Sample collection and RT-qPCR diagnosis

In mid-January 2021, routine genomic surveillance in the Central Laboratory of Health of the Bahia state (LACEN-BA), started an extensive screening of COVID-19 patients and their contacts reporting a travel history to/from the Amazonas state, resulting in eleven suspected SARS-CoV-2, P.1 infections.

Viral RNA was extracted from nasopharyngeal swabs using an automated protocol on the King Fisher platform using the MagMax Kit (Thermofisher Scientific) and tested for SARS-CoV-2 by multiplex real-time PCR assays: i) Allplex 2019-nCoV Assay (Seegene) targeting the

envelope (E), the RNA dependent RNA polymerase (RdRp) and the N genes and ii) the Charité: SARS-CoV2 (E/RP) assay (Biomanguinhos) targeting the E gene, supplied by the Brazilian Ministry of Health. Samples were selected based on the Ct value ≤ 32. Associated epidemiological data, such as symptoms, travel history and municipality of residency, were collected from medical records accompanying the collected samples provided by LACEN-BA.

## Library preparation

Samples with RT-PCR–positive were submitted to the Ion GeneStudio S5 Plus System (Life Technologies, USA) to viral genomic amplification and subsequent sequencing according to the manufacturer's instructions.

## Generation of consensus sequences

Raw files were basecalled using Guppy v3.4.5 and barcode demultiplexing was performed using qcat v.1.1.0. Consensus sequences were generated by de novo assembling using Genome Detective (https://www.genomedetective.com/) [10].

## Phylogenetic analysis

We explored the genetic relationship of the newly sequenced P.1 genomes to those of other isolates by phylogenetic inference. To do so, we combined the eleven new isolates (Accession numbers EPI_ISL_1067728—EPI_ISL_1067738) with all Brazilian SARS-CoV-2 (n = 1663) genomes, including recently released P.1 genomes [4,5] available on GISAID (https://www.gisaid.org/) up to February 21st, 2021 (a fully annotated tree can be found in the S1 Fig). Only genomes >29,000bp and <1% of ambiguities were considered (n = 1663). Sequences were aligned using MAFFT v.7 [11] and submitted to IQ-TREE 2 for maximum likelihood (ML) phylogenetic analysis [12]. The ML phylogenetic tree was inferred under the GTR+F+I+G4 nucleotide substitution model as selected by the ModelFinder application and the branch support was assessed by the approximate likelihood-ratio test based on the bootstrap and the Shimodaira–Hasegawa-like procedure (SH-aLRT) with 1,000 replicates. Lineages assessment was conducted using Phylogenetic Assignment of Named Global Outbreak Lineages tool available at https://github.com/hCoV-2019/pangolin [1].

## Results

In mid-January 2021, samples from (clinically) suspected cases and their contact reporting a travel history to/from the Amazonas state were screened at the Central Laboratory of Health of the Bahia state (LACEN-BA). A total of 11 RT-qPCR positive samples were screened. All samples tested, contained sufficient DNA (≥2ng/μL) to proceed to library preparation. For those positive samples, PCR cycle threshold (Ct) values for common target (E gene) ranged from ≈16 to ≈30 (Table 1). We next subjected the qRT-PCR–positive samples to viral genomic amplification and sequencing using the Ion GeneStudio™ S5 Plus Ion Torrent (Life Technologies, USA), according to the manufacturer's instructions. A total of 13,234,814 mapped reads were obtained, resulting in a sequencing mean depth of 4824 and a coverage of 98% (Table 1). The new whole genome sequences generated were assigned, according to the Pangolin [1] lineage classification, as the recently identified P.1 Brazilian VOC. All patients had recently recorded travel from the city of Manaus in Amazonas back to the Bahia state. Four patients were from the same family and the others had no other known connection between them. The data from the eleven patients are described in (Table 1).

**Table 1. Information from patients with travel history from Manaus, Amazonas to state of Bahia.**

| GISAID ID | ID | Colection Date | Sex | Age | Municipality | Symptoms | Protocol | Cycle threshold (ct) | Reads | Coverage | Depth of Coverage |
|---|---|---|---|---|---|---|---|---|---|---|---|
| EPI_ISL_1067737 | BA49[1] | 2021-01-26 | M | 7 | Salvador | Assymptomatic | Allplex 2019-nCoV Assay (Seegene) | CI -16.8 E—18.1 N—20.8 RdRp—26.5 | 1659219 | 99.8 | 10179.3 |
| EPI_ISL_1067730 | BA50[1] | 2021-01-26 | F | 31 | Salvador | Sore throat, cough, fever | Allplex 2019-nCoV Assay (Seegene) | CI -28.7 E—25.7 N—25.6 RdRp—28.4 | 594545 | 99.8 | 3706.8 |
| EPI_ISL_1067738 | BA51[1] | 2021-01-26 | M | 66 | Salvador | Sore throat, cough, fever | Allplex 2019-nCoV Assay (Seegene) | CI -27.0 E—24.9 N—23.8 RdRp—25.6 | 412507 | 82.2 | 2869 |
| EPI_ISL_1067731 | BA52[1] | 2021-01-26 | F | 37 | Salvador | Sore throat, cough, fever | Allplex 2019-nCoV Assay (Seegene) | CI -27.5 E -24.5 N—25.5 RdRp -26.8 | 1047634 | 99.2 | 6779 |
| EPI_ISL_1067735 | BA 56[1] | 2021-01-16 | M | 37 | Salvador | Sore throat, cough, fever | Allplex 2019-nCoV Assay (Seegene) | CI -25.2 E -24.9 N—25.7 RdRp -26.6 | 862894 | 99.8 | 5438.3 |
| EPI_ISL_1067729 | BA 54 | 2021-01-23 | F | 56 | Irecê | Sore throat, cough, fever, fatigue, dyspnea, coriza, dyspnea, diarrhea, respiratory distress, $O_2$ saturation = 92% | Allplex 2019-nCoV Assay (Seegene) | CI -15.4 E -16.9 N—16.8 RdRp -27.2 | 625085 | 99.8 | 3982.5 |
| EPI_ISL_1067734 | BA 55 | 2021-01-23 | M | 52 | João Dourado | Fever, headache | Allplex 2019-nCoV Assay (Seegene) | CI -27.0 E -22.3 N—23.7 RdRp -24.4 | 1848679 | 99.8 | 11278.1 |
| EPI_ISL_1067733 | BA 53 | 2021-01-23 | M | 19 | João Dourado | Assymptomatic | Charité: SARS-CoV2 (E/RP) (Bio Manguinhos) | E—22.9 RP—26.6 | 1131609 | 99.8 | 6979.9 |
| EPI_ISL_1067736 | BA57 | 2021-01-23 | M | 42 | Salvador | Fever, headache | Allplex 2019-nCoV Assay (Seegene) | CI -25.5 E -17.2 N—18.9 RdRp -19.2 | 1416051 | 99.8 | 8970.8 |
| EPI_ISL_1067728 | BA58 | 2021-01-08 | F | 53 | Salvador | Fever, headache | Allplex 2019-nCoV Assay (Seegene) | CI -27.8 E -29.0 N—29.7 RdRp -30.0 | 2343392 | 98 | 14635.7 |
| EPI_ISL_1067732 | BA59 | 2021-01-18 | M | 57 | Salvador | Fever, headache | Allplex 2019-nCoV Assay (Seegene) | CI -26.7 E -30.0 N—30.0 RdRp -30.0 | 1293199 | 99.8 | 8105.2 |

[1]Patients from the same familiy

Phylogenetic analysis strongly supported placement of the isolates from Bahia within the Brazilian P.1 clade (Fig 1A) (Bootstrap = 1.0, SH-aLTR = 1.0).

Our results further highlighted that most of our new isolates formed two well supported monophyletic clusters (Fig 1C). One of those clusters (**Cluster i**), included samples from the

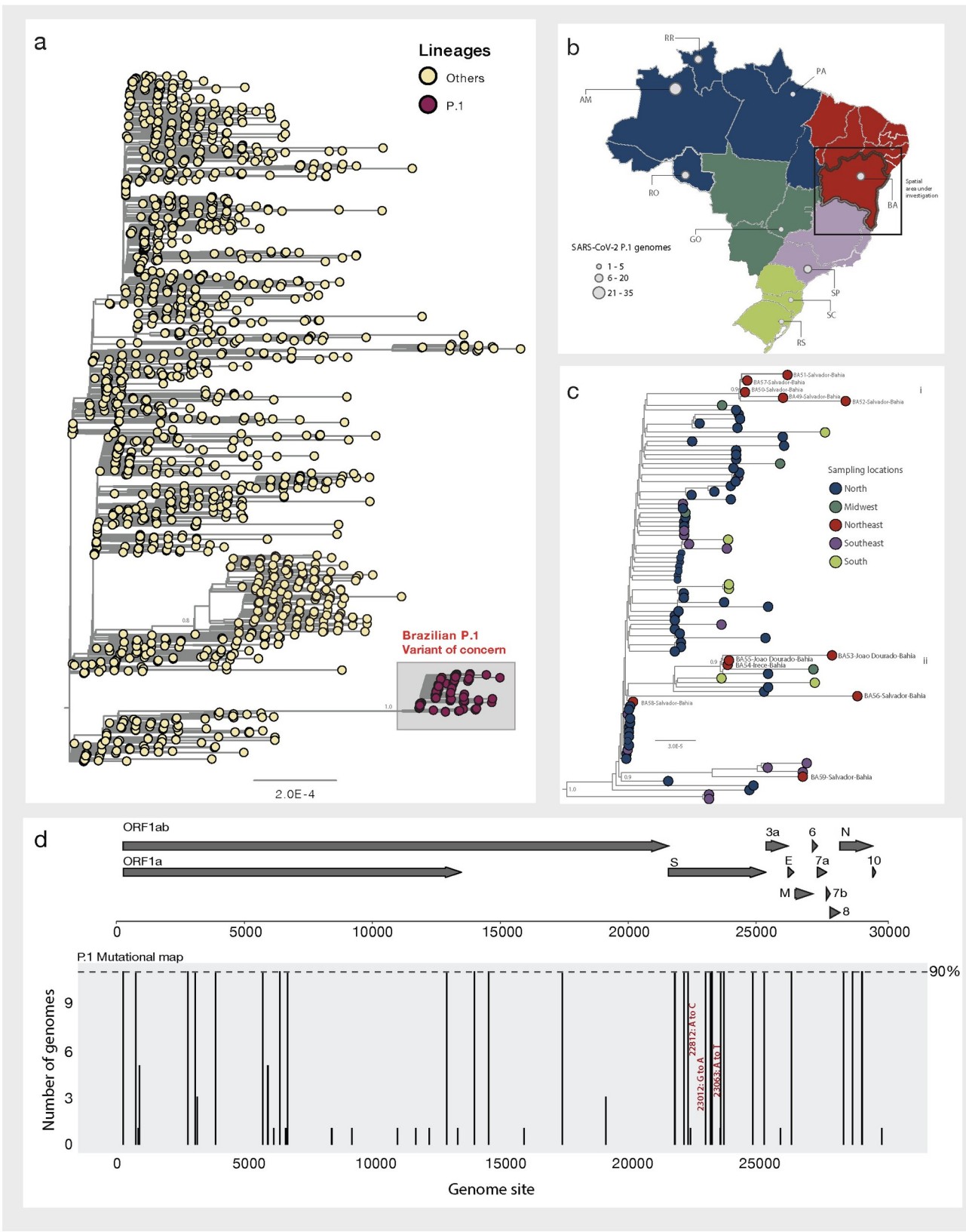

**Fig 1. Genomic detection of the SARS-CoV-2 P.1 variant of concern in Bahia state, Northeast Brazil.** a) Maximum likelihood (ML) phylogenetic tree including the 11 new isolates obtained in this study plus n = 1663 SARS-CoV-2 Brazilian strains collected up to February 21st, 2021. b) Map of Brazil

(generated in R using the "*get_brmap*" package available at: https://rdrr.io/cran/brazilmaps/man/get_brmap.html) showing the number of P.1 SARS-CoV-2 sequences available by region and state (updated up to when this surveillance was done—February 21st, 2021). North region (blue): AM = Amazonas state, RR = Roraima state and RO = Rondônia state; Midwest region (dark green): GO = Goiás state; Southeast region (lilac): SP = São Paulo and South region: SC (light green): Santa Catarina state and RS = Rio Grande do Sul state and Northeast region (red): BA = Bahia state. c) Representation of the zoom of the Brazilian P.1 clade. Branch support (Bootstrap = 1.0, SH-aLTR = 1.0) is shown at key nodes. d) Variant maps of the P.1 lineage-defining-mutations were mapped against the SARS-CoV-2 genome structure. Most common mutations defined as mutations present in more than 90% of the genomes in that group. Mutations of international concern: i) K417T (22812A>C); ii) E484K (23012G>A) and iii) N501Y (23063A>T), among the RDB domain are highlighted in red.

capital city of Salvador, isolated from individuals from the same family (BA49, 50, 51 and 52) plus the sample BA57 (Bootstrap = 1.0, SH-aLTR = 1.0). And a second cluster (**Cluster ii**) which included samples BA53, 54 and 55 that were isolated from patients from the municipalities of João Dourado and Irecê (Bootstrap = 1.0, SH-aLTR = 1.0). Moreover, samples BA56 and BA58 and BA59 appeared to be interspersed among P.1 strains from the North and Southeast Brazil. By combining all the P.1 strains already available from distinct Brazilian regions (**Fig 1B and 1C**) our analysis further revealed that this variant was already detected in the majority of Brazilian regions (starting from the North to Midwest, Northeast, Southeast and South Brazil) highlighting the high connectivity of the country and reinforcing the need of active monitoring to follow the local real-time spread of this new variant of international concern. Finally, among the eleven new genomes, we also identified lineage-specific mutations, including the ones of international concern among the RDB domain: K417T (22812A>C), E484K (23012G>A) and the N501Y (23063A>T) (**Fig 1D**).

## Discussion

This report describes the early detection of the SARS-CoV-2 P.1 variant in the Northeast of Brazil (Bahia state) and provides evidence regarding the P.1 circulation across all Brazilian macro-regions that would have occurred within the past two months. The P.1 variant is now known to have emerged in the Amazonas state, but was first detected in travelers arriving in Japan from Brazil. The latter is a testimony for the currently scarce genomic surveillance in Brazil, which has failed to detect the variant before it became a public health emergency in the Amazonas state. Moreover, the eleven individuals here described to carry the P.1 variant in the Bahia state were only detected after a unique active screening initiative that focused on travelers and their recent contacts. This case study should serve as an example of the effectiveness of active surveillance to monitor the importation of genetic variants of importance for public health. Since such initiatives are not only critical to monitor the ongoing spread of known variants but are also necessary to detect the possible emergence of new ones in the near future, continual and revamped investment is needed and will be necessary for adequate public health policy in Brazil.

## Supporting information

**S1 Table. GISAID acknowledgment table.**
(PDF)

**S1 Fig. Fully annotated Maximum likelihood (ML) phylogenetic tree including the 11 new isolates obtained in this study plus n = 1663 SARS-CoV-2 Brazilian strains collected up to February 21st, 2021.**
(TIF)

## Acknowledgments

We thank the "*Centro e Informações Estratégicas de Vigilância em Saúde*" (CIEVS) in the municipality of Salvador, Bahia and Pan American Health Organization PAHO/WHO. We also would like to thank all the authors who have kindly deposited and shared genome data on GISAID. A table with genome sequence acknowledgments can be found in **S1 Table.**

## Author Contributions

**Conceptualization:** Marta Giovanetti, Felicidade Pereira, Arabela Leal.

**Data curation:** Stephane Tosta, Marta Giovanetti, Vagner Fonseca.

**Formal analysis:** Stephane Tosta, Marta Giovanetti, Vanessa Brandão Nardy, Luciana Reboredo de Oliveira da Silva, Marcela Kelly Astete Gómez, Jaqueline Gomes Lima, Cristiane Wanderley Cardoso, Vagner Fonseca.

**Funding acquisition:** Tarcisio Oliveira Silva.

**Investigation:** Stephane Tosta, Marta Giovanetti, Vanessa Brandão Nardy, Luciana Reboredo de Oliveira da Silva, Marcela Kelly Astete Gómez, Jaqueline Gomes Lima, Cristiane Wanderley Cardoso, Tarcisio Oliveira Silva, Pedro Henrique Presta Dia, Vagner Fonseca, Felicidade Pereira, Arabela Leal.

**Methodology:** Stephane Tosta, Marta Giovanetti, Vanessa Brandão Nardy, Luciana Reboredo de Oliveira da Silva, Marcela Kelly Astete Gómez, Jaqueline Gomes Lima, Cristiane Wanderley Cardoso, Marcia São Pedro Leal de Souza, Pedro Henrique Presta Dia, Vagner Fonseca, Felicidade Pereira, Arabela Leal.

**Project administration:** Felicidade Pereira, Arabela Leal.

**Resources:** Cristiane Wanderley Cardoso, Tarcisio Oliveira Silva, Pedro Henrique Presta Dia, Felicidade Pereira.

**Software:** Marta Giovanetti.

**Supervision:** Marta Giovanetti, Tulio de Oliveira, José Lourenço, Felicidade Pereira.

**Validation:** Marta Giovanetti, Marcia São Pedro Leal de Souza, Tulio de Oliveira, José Lourenço, Felicidade Pereira.

**Visualization:** Marta Giovanetti, Marcia São Pedro Leal de Souza, Vagner Fonseca, José Lourenço, Luiz Carlos Junior Alcantara, Felicidade Pereira, Arabela Leal.

**Writing – original draft:** Stephane Tosta, Marta Giovanetti.

**Writing – review & editing:** Marta Giovanetti, Vagner Fonseca, Tulio de Oliveira, José Lourenço, Luiz Carlos Junior Alcantara, Felicidade Pereira, Arabela Leal.

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
