## [Decision Letter · Decision Letter 0]

7 May 2021

Dear Dr. Giovanetti,

Thank you very much for submitting your manuscript "Short Report: Early genomic detection of SARS-CoV-2 P.1 variant in Northeast Brazil" for consideration at PLOS Neglected Tropical Diseases. As with all papers reviewed by the journal, your manuscript was reviewed by members of the editorial board and by several independent reviewers. In light of the reviews (below this email), we would like to invite the resubmission of a significantly-revised version that takes into account the reviewers' comments. 

We cannot make any decision about publication until we have seen the revised manuscript and your response to the reviewers' comments. Your revised manuscript is also likely to be sent to reviewers for further evaluation.

Sincerely,

Colleen B Jonsson, PhD

Associate Editor

Emma Wise

Deputy Editor

Reviewer's Responses to Questions

**Key Review Criteria Required for Acceptance?**

**Methods**

-Are the objectives of the study clearly articulated with a clear testable hypothesis stated?

-Is the study design appropriate to address the stated objectives?

-Is the population clearly described and appropriate for the hypothesis being tested?

-Is the sample size sufficient to ensure adequate power to address the hypothesis being tested?

-Were correct statistical analysis used to support conclusions?

-Are there concerns about ethical or regulatory requirements being met?

Reviewer #1: (No Response)

Reviewer #2: na

**Results**

-Does the analysis presented match the analysis plan?

-Are the results clearly and completely presented?

-Are the figures (Tables, Images) of sufficient quality for clarity?

Reviewer #1: (No Response)

Reviewer #2: The data is clearly presented

**Conclusions**

-Are the conclusions supported by the data presented?

-Are the limitations of analysis clearly described?

-Do the authors discuss how these data can be helpful to advance our understanding of the topic under study?

-Is public health relevance addressed?

Reviewer #1: (No Response)

Reviewer #2: (No Response)

**Editorial and Data Presentation Modifications?**

Reviewer #1: (No Response)

Reviewer #2: The manuscript “Early genomic detection of SARS-CoV-2 P.1 variant in Northeast Brazil” provides a brief report resulting from the analyses of 11 viral genomes. While the dataset is limited, it comes from an epidemiologically interesting region at an important time in the dispersal of the P.1 variant. As such, the manuscript provides information that will be of general interest. The message is simple and clearly presented by the authors. Some minor comments for the authors consideration follow.

1. Ln 78. The authors state that eleven suspected P1 infections resulted from the massive screening program. It is a little unclear exactly what this means. Were only 11 positive samples detected through this screening (I suspect this is not what is meant)? It might be clearer if the number of tests conducted and the number of positive PCR tests were provided. If more than 11 samples were positive, how were these particular samples selected? 

2. Ln 129. What was the rationale for using two PCR assays on each sample?

3. Ln 136. Did all four family members travel, or was there evidence of within household spread?

4. It would be of interest to know the spectrum of disease experienced by these individuals.

Editorial suggestions

1. Ln 133. “generated where assigned” should be “generated were assigned”

2. Ln 135. “recently record travel” should be “recently recorded travel”

3. Ln 136. “back into the” should be “back to the”

4. Ln 136-137. “and the other others had” should be “and the others had”

**Summary and General Comments**

Reviewer #1: Tosta et.al. present the manuscript entitled “Early genomic detection of SARS-CoV-2 P.1 variant in Northeast Brazil” described a pilot study in the northern region of brazil to recover the 11 SARS-COV2 genomes contain P.1 variant, which is of the active public health concern strain. As P1 variants is the important for public health on the active COVID19 pandemic right now, the manuscript will be interested in researchers of the closely related filed. However, the manuscript needs to be improved, especially in the method section. The inconsistency between method and result needs to be clarified. 

Comments

1. The P.1 variant need to be defined in detail at first in the Author summary section.

2. Please provide version of the software used in the study: Guppy, qcat, MAFFT, IQ_TREE

3. As Phylogenetic tree analysis is very important result of the study, I recommend the author give the reader more details how to construct the tree such as what is the parameter used, what is the evolutionary model used. Boot strapping is highly recommended to perform and report as confident of the tree result. 

4. It is very inconsistency in the method and result! The authors mentioned that they use Oxford Nanopore sequencing of PCR product from ARCTIC primer set, but the first paragraph of the result mentioned that Ion torrent sequencer was used. The authors need to clarify experimental procedures in a better detail. 

5. Figure 1A need to be improved. The annotation of different clades based on known linages need to be reported on the figure. Readers will not gain any knowledge from the figure without the annotation of leakages. By using “Others” is meaningless. Standard tree file format such as newick format need to be shared as a supplementary for the readers.

6. The label on y-axis of Figure 1D is missing.

7. Raw sequence data need to be deposited and shared for the research community in a public database such as NCBI SRA database. 

8. For reproducible and transparency of the result, I recommended the authors share the computational notebook or chunk of code used in the study.

Reviewer #2: (No Response)

PLOS authors have the option to publish the peer review history of their article (what does this mean?). If published, this will include your full peer review and any attached files.

Reviewer #1: No

Reviewer #2: Yes: Richard J Webby
---

## [Decision Letter · Decision Letter 1]

23 Jun 2021

Dear Dr. Giovanetti,

We are pleased to inform you that your manuscript 'Short Report: Early genomic detection of SARS-CoV-2 P.1 variant in Northeast Brazil' has been provisionally accepted for publication in PLOS Neglected Tropical Diseases.

Best regards,

Colleen B Jonsson, PhD

Associate Editor

Emma Wise

Deputy Editor

Reviewer's Responses to Questions

**Key Review Criteria Required for Acceptance?**

**Methods**

-Are the objectives of the study clearly articulated with a clear testable hypothesis stated?

-Is the study design appropriate to address the stated objectives?

-Is the population clearly described and appropriate for the hypothesis being tested?

-Is the sample size sufficient to ensure adequate power to address the hypothesis being tested?

-Were correct statistical analysis used to support conclusions?

-Are there concerns about ethical or regulatory requirements being met?

Reviewer #1: (No Response)

Reviewer #2: my concerns have been addressed

**Results**

-Does the analysis presented match the analysis plan?

-Are the results clearly and completely presented?

-Are the figures (Tables, Images) of sufficient quality for clarity?

Reviewer #1: (No Response)

Reviewer #2: my concerns have been addressed

**Conclusions**

-Are the conclusions supported by the data presented?

-Are the limitations of analysis clearly described?

-Do the authors discuss how these data can be helpful to advance our understanding of the topic under study?

-Is public health relevance addressed?

Reviewer #1: (No Response)

Reviewer #2: my concerns have been addressed

**Editorial and Data Presentation Modifications?**

Reviewer #1: (No Response)

Reviewer #2: (No Response)

**Summary and General Comments**

Reviewer #1: The author improved the manuscript and addressed my comments.

Reviewer #2: my concerns have been addressed

PLOS authors have the option to publish the peer review history of their article (what does this mean?). If published, this will include your full peer review and any attached files.

Reviewer #1: No

Reviewer #2: **Yes: **Richard Webby

---

## [Editor Report · Acceptance letter]

15 Jul 2021

Dear Dr. Giovanetti,

We are delighted to inform you that your manuscript, "Short Report: Early genomic detection of SARS-CoV-2 P.1 variant in Northeast Brazil," has been formally accepted for publication in PLOS Neglected Tropical Diseases.

Best regards,

Shaden Kamhawi

co-Editor-in-Chief

Paul Brindley

co-Editor-in-Chief
